# Impact of the COVID-19 Pandemic on the Incidence of Suicidal Behaviors: A Retrospective Analysis of Integrated Electronic Health Records in a Population of 7.5 Million

**DOI:** 10.3390/ijerph192114364

**Published:** 2022-11-02

**Authors:** Damià Valero-Bover, Marc Fradera, Gerard Carot-Sans, Isabel Parra, Jordi Piera-Jiménez, Caridad Pontes, Diego Palao

**Affiliations:** 1Catalan Health Service, Travessera de les Corts, 131-159 Edifici Olímpia, 08028 Barcelona, Spain; 2Digitalization for the Sustainability of the Healthcare System DS3—IDIBELL, Gran via de l’Hospitalet Número 199-203, 08908 L’Hospitalet de Llobregat, Spain; 3Department of Mental Health, University Hospital ParcTaulí, Parc Tauí, 08208 Sabadell, Spain; 4Unitat Mixta de Neurociència Traslacional I3PT-INc-UAB, Institut d’Investigació i Innovació Parc Taulí I3PT, Centro de Investigación Biomédica en Red de Salud Mental, Edifici Santa Fe, 08208 Sabadell, Spain; 5Facultat de Medicina, Universitat Autònoma de Barcelona, Campus de la Universitat Autònoma de Barcelona, 08193 Bellaterra, Spain; 6Faculty of Informatics, Multimedia and Telecommunications, Universitat Oberta de Catalunya, Rambla de Poblenou 156, 08018 Barcelona, Spain; 7Department of Pharmacology, Therapeutics and Toxicology, Universitat Autònoma de Barcelona, Edifici M Campus de la UAB, 08193 Bellaterra, Spain

**Keywords:** suicide, COVID-19 pandemic, mental health

## Abstract

The COVID-19 pandemic has caused remarkable psychological overwhelming and an increase in stressors that may trigger suicidal behaviors. However, its impact on the rate of suicidal behaviors has been poorly reported. We conducted a population-based retrospective analysis of all suicidal behaviors attended in healthcare centers of Catalonia (northeast Spain; 7.5 million inhabitants) between January 2017 and June 2022 (secondary use of data routinely reported to central suicide and diagnosis registries). We retrieved data from this period, including an assessment of suicide risk and individuals’ socioeconomic as well as clinical characteristics. Data were summarized yearly and for the periods before and after the onset of the COVID-19 pandemic in Spain in March 2020. The analysis included 26,458 episodes of suicidal behavior (21,920 individuals); of these, 16,414 (62.0%) were suicide attempts. The monthly moving average ranged between 300 and 400 episodes until July 2020, and progressively increased to over 600 episodes monthly. In the postpandemic period, suicidal ideation increased at the expense of suicidal attempts. Cases showed a lower suicide risk; the percentage of females and younger individuals increased, whereas the prevalence of classical risk factors, such as living alone, lacking a family network, and a history of psychiatric diagnosis, decreased. In summary, suicidal behaviors have increased during the COVID-19 pandemic, with more episodes of suicidal ideation without attempts in addition to younger and lower risk profiles.

## 1. Introduction

Soon after the first case of infection by severe acute respiratory syndrome coronavirus 2 (SARS-CoV-2) in December 2019, the virus spread rapidly all around the globe, leading to an unprecedented global health crisis. Aside from the direct effects of coronavirus disease 2019 (COVID-19), the pandemic has had other important impacts derived from the measures implemented for limiting the spread of SARS-CoV-2. These measures, primarily based on quarantines (either general or targeting cases and contacts), social distancing, and the banning of gatherings [1], have altered social interactions, with potential impacts on mental health, particularly among individuals with previous psychiatric pathologies [2,3,4,5]. Aside from the stressors directly derived from social distancing, other factors, such as the uncertainty associated with the pandemic scenario, the impossibility of accompanying loved ones during hospital stays or end-of-life stages, or the financial crises (including unemployment) experienced in many countries, have contributed to psychological overwhelming in many cases [6]. Likewise, successive waves of COVID-19 translated into periods of the intensification and relaxing of social restrictions, according to infectious transmission rates; such intermittent distortions of social interaction have caused learned helplessness behaviors, increasing the risk of depressive disorders [7]. Irrespective of the cause of declining mental health, various authors have reported the increasing use of mental health services during the pandemic [3,4].

Being overwhelmed, experiencing disaster, and a sense of isolation are among the triggers of suicidal behaviors, particularly among individuals with underlying mental disorders, such as depression, schizophrenia, or alcohol abuse, among others [8]. In 2019, before the onset of the COVID-19 pandemic, the number of deaths due to suicide reported by the WHO amounted to over 700,000 cases worldwide, with suicide being the fourth leading cause of death among 15 to 29 year olds [9]. Although the report showed a decline in suicide rates in Europe between 2000 and 2019, the impact of the COVID-19 pandemic on mental health is likely to reverse this trend, and figures in some countries suggest that the death toll due to suicide could be even greater than that directly attributed to SARS-CoV-2 infection [10,11].

To date, various authors have reported changes in the incidence and lethality of suicidal behaviors associated with the COVID-19 pandemic [12,13,14,15]. However, the lack of integrated health information in many countries limits the number of population-based analyses of the shift in suicide rates before and after the COVID-19 pandemic. Likewise, little is known about whether the profiles of individuals who attempt suicide before and after the pandemic onset are changing. We hypothesize that the COVID-19 pandemic has influenced not only the incidence of suicidal behaviors but also the clinical and sociodemographic profiles of individuals experiencing suicidal behavior events. Therefore, the objective of this retrospective analysis of a population of 7.5 million was to assess the changes in suicidal behaviors and individual profiles before and during the first two years of the COVID-19 pandemic by retrieving data from the dataset of the Suicide Risk Code [16], a secondary prevention program implemented in 2014 in Catalonia (northeast Spain) to reduce mortality due to suicide. 

## 2. Materials and Methods

### 2.1. Study Design and Setting

This was a retrospective analysis of electronic health records of Catalonia, an area with 7.5 million people in northeast Spain. The Catalan Health Service (CHS) provides public, universal healthcare to the entire population of Catalonia. Exclusive private care is scarcely used, particularly in emergency settings; hence, the CHS holds the majority of the health information of Catalan individuals, and nearly all cases of admissions to emergency rooms. Although information systems are fragmented into various datasets, by hospital, primary care, and mental health, among others, the unique identification numbers used for insurance purposes allow for the integration of information across the entire system. 

In Catalonia, the COVID-19 pandemic started in early March 2020, and a state of emergency was declared by 14 March 2020, with strict lockdown measures that lasted until 21 June 2020. Measures to contain viral transmission, such as curfews, limited mobility across the territory, social restrictions in public spaces, and mandatory face masks, were applied in the following two years with different degrees of intensity according to the needs of six successive epidemic waves. Measures taken during the first lockdown period reduced healthcare activity, which later recovered. Background information regarding mental health outpatient activity and hospital activity during this period is provided in Appendix A.

In 2014, the CHS launched the Suicide Risk Code, a secondary prevention program aimed at reducing mortality due to suicide, increasing survival among individuals attended to because of suicidal behaviors, and preventing repeated suicidal attempts in patients with risk factors [16]. One of the components of the Suicide Risk Code was a specific registry for recording all cases of suicidal behavior attended to in CHS health centers and collecting information regarding episodes, key clinical and social conditions, and care pathways followed after episodes. The Suicide Risk Code was progressively implemented until it covered the entire territory of Catalonia in 2016. Since then, all individuals with a suspected risk of suicide that visited any healthcare center of the CHS are referred to the closest psychiatry ward for a comprehensive assessment and are registered in the Suicide Risk Code dataset, regardless of the presence of a suicide attempt. To prevent biases due to the incomplete implementation of the program, for this analysis we screened the Suicide Risk Code dataset for episodes that occurred from 01 January 2017 to the end of the investigated period in 30 June 2022. 

### 2.2. Ethical Conduct of the Study

The study protocol was approved by the independent ethics committee of the Consortium Corporació Sanitària Parc Taulí of Sabadell, which waived the collection of informed consent for the secondary use of health data collected during routine care (approval code 20225075 of 13 September 2022). The study was conducted according to the General Data Protection Regulation 2016/679 on data protection and privacy for all individuals within the European Union in addition to the local regulatory framework regarding data protection.

### 2.3. Variables and Data Sources

The characteristics of suicidal behavior episodes and risk factors recorded at the times of episodes were retrieved from the Suicide Risk Code dataset. The type of suicidal behavior episode was classified as either active suicidal ideation without an attempt or a suicide attempt, defined as engaging in self-directed injury with the specific intent to die. In patients with a suicide attempt, the method used in the attempt was recorded. The following variables were recorded during the visit that triggered an entry in the Suicide Risk Code registry: family history of suicide, severe or painful systemic disease, abuse of alcohol or drugs, recent economic or personal stressors, lack of family or social network, living alone, and recent feelings of hopelessness. We also collected the suicide risk level, rated based on the 6-item suicidality module of the Mini International Neuropsychiatric Interview (MINI). The MINI scale has been validated in relation to the Structured Clinical Interview for DSM-III-R and the Composite International Diagnostic Interview (CIDI) for the International Statistical Classification of Disease (ICD)-10 [17]. The suicidality score ranges from 0 to 33, and allows for the grouping of risk into low (0–5 points), moderate (6–9 points), and high (≥10 points) [17]. The list of variables included in the Suicide Risk Code dataset and their definitions are provided in Appendix A.

The CHS identification numbers were used to retrieve demographic characteristics (i.e., age and sex) and psychiatric disorders diagnosed before or up to one month following the suicidal behavior episodes, including the following: schizophrenia and schizoaffective disorders, eating disorders, anxiety, depression (including recurrent or persistent mood (affective) disorders), disorders of personality and behavior, sleep disorders, Alzheimer’s disease, and psychoactive substance use. The International Classification of Diseases version 10—Clinical Modification (ICD-10-CM) codes are listed in Appendix A. 

Finally, we collected data on socioeconomic status based on the pharmaceutical co-payment classification of the CHS, which stratifies the population into four socioeconomic groups based on the pharmaceutical co-payment: very low (i.e., individuals with minimum integration income, unemployment allowance, and unemployment benefit, in addition to those on leave due to a work-related accident or professional disease, persons with severe disability, and other highly vulnerable groups), low (annual income of <EUR 18,000), moderate (annual income of EUR 18,000 to EUR 100,000), and high (annual income of >EUR 100,000). During the 2020–2021 period, the criteria for very low and low groups changed to widen the coverage of the very low group; therefore, for the purpose of this work, we grouped these two categories.

### 2.4. Analysis

The progression of suicidal behavior episodes throughout the investigated period was plotted in terms of monthly incidence (observed and moving average). The time series of the observed incidences were decomposed into trend, seasonal, and random components. The moving average, used to smooth out short-term fluctuations in the time series and to highlight longer-term trends and cycles, was calculated as an average value of data points covering 12 months, using a symmetric window with equal weights. The seasonal component was computed by removing the trend component from the time series and averaging, for each time unit, over all periods. The random component was determined by removing the trend and seasonal components from the original time series. The monthly incidence was determined for the entire sample and for age, sex, and socioeconomic groups of interest. Data on the Catalan population were retrieved from the Statistical Institute of Catalonia for incidence estimates [18]. The characteristics of episodes and the clinical as well as demographic profiles of individuals were described with frequency and percentage over available data. Continuous variables were described by using the mean and standard deviation (SD). Episodes’ characteristics and individual profiles were summarized for each natural year of the investigated period. Additionally, we estimated the percentage of each risk factor and episode characteristic before and after the onset of the COVID-19 pandemic in Catalonia, in March 2020, considering natural months as units. Owing to this population-based approach and focus on episodes (rather than individuals) for the primary endpoint, we conducted a descriptive analysis, with no hypothesis testing. All analyses were conducted in R version 4.0.4 [19]; plots were built using package ggplot version 3.3.3 [20]. 

## 3. Results

### 3.1. Incidence of Suicidal Behavior

Between January 2017 and June 2022, the Suicide Risk Code registered 26,458 episodes of suicidal behavior in 21,920 individuals. The monthly rate of suicidal behavior events dropped by to an extreme extent during the two months of nationwide lockdown, and the overall trend remarkably increased afterwards (Figure 1a). Overall, the rate of suicidal behaviors per 100,000 inhabitants per month ranged from 3.06 to 6.17 before the onset of the pandemic (2017–2019, both included) and 7.43 to 9.42 within the last six months of the observation period (January-June 2022). The increase in incidence was more pervasive among females (Figure 1a) and minors (Figure 1b). 

The subgroup analysis according to gender and age confirmed females under 18 years old as the primary contributors to the overall increase in suicidal behaviors in the general population (Figure 2a). The analysis according to socioeconomic status showed increases in all groups, with a persistently higher incidence in the low-socioeconomic-status group (Figure 2b). The larger increase in incidence among female and minors was consistent across all of the socioeconomic groups (Appendix A).

### 3.2. Risk Factors and Characteristics of Suicidal Behaviors

Table 1 summarizes the characteristics of the suicidal behavior episodes through the investigated period. Compared with the three years before the onset of the COVID-19 pandemic, episodes reported during the pandemic were less frequently attempts, and individuals showed a lower suicidal risk in the MINI assessment at the times of attempts. The mean (SD) MINI scores were 13.5 (8.9), 12.9 (8.4), and 12.0 (8.3) for 2017, 2018, and 2019, respectively, and 11.6 (8.4), 11.1 (8.1), and 11.1 (8.0) for 2020, 2021, and the first semester of 2022, respectively. 

The analysis of the underlying social and clinical characteristics of the cases throughout the investigated period revealed an increase in the percentage of younger individuals and females in the years following the onset of the pandemic (Table 2). Additionally, individuals with suicidal behaviors within the years following the onset of the pandemic were less likely to present with classical risk factors for suicide attempts, such as living alone, lacking social or family support, or having an underlying mental disorder.

The average values before and after the onset of the pandemic confirmed the trends observed in the yearly analysis (Figure 3). In both periods, self-poisoning with liquid or solid substances remained the leading method among those with attempts, with no remarkable changes observed regarding the frequency of each method between periods (Figure 3a). However, after the onset of the pandemic, suicidal behavior episodes were more frequently ideations without suicide attempts. The mean number of suicidal behavior episodes per case was similar before (1.3 (SD: 0.9)) and after (1.4 (SD: 1.0)) the onset of the pandemic. 

The most remarkable change in individual profiles was the lower frequency of mental disorders after the onset of the pandemic, evidenced in all psychiatric diagnoses, particularly in regard to schizophrenia and schizoaffective disorders (Figure 3b). Depression, anxiety, and drug use were consistently the most prevalent mental health disorders. 

## 4. Discussion

In this population-based, retrospective analysis of suicidal behaviors among a population of 7.5 million, 21,920 cases of suicidal behavior were identified through the population-based Suicide Risk Code registry between January 2017 and June 2022. In this study population, we found that the COVID-19 crisis led to a remarkable increase in the rate of suicidal behavior episodes, primarily explained by an increase in incidence among young females. The turning point in the trend of suicidal behaviors was pervasive after June 2020, four months after the onset of the outbreak in our country. We also found changes in the profiles of individuals with suicidal behaviors, who presented with a lower prevalence of classical risk factors for suicide and suicidal behaviors, such as alcohol/drug abuse or underlying psychiatric disorders.

The overall increase in suicidal behaviors observed in our population is consistent with that described by other authors. Reif-Leonhard et al. found a lower rate of suicide attempts reported at the Frankfurt program to prevent suicides in 2020 compared to 2019 [13]. Similarly, demand for the suicide prevention helpline in the Netherlands decreased within the first months of the pandemic and started an increasing trend approximately four months after the first case in the country [12]. Finally, Yoshioka et al. observed that the decreasing trend in suicides experienced in Japan during the 2016–2019 period was reverted in the overall 2020–2021 period, although a delay of several months in the rough observed cases was reported [15]. This delay was in line with our analysis, which revealed a lag time of almost 4 months after the declaration of the state of emergency and population lockdown. Interestingly, our analysis showed that the strict lockdown period, an important stressor associated with an increased number of anxiety episodes and psychiatric consultations in our area [7,21], was followed by a drop in suicidal behavior events. This phenomenon was also observed in a city-based study in our area [22]. Similarly, in the Netherlands, decreasing demand for the suicide prevention helpline was also observed in a nationwide analysis right after the general lockdown, although eight months later a partial lockdown prompted a peak in demand for the same service [12]. This delay in the change in suicide rates or suicidal behaviors has also been observed in other natural disasters [23,24]. In the case of lockdowns associated with the COVID-19 pandemic, the protective effect of pulling together might have also contributed to the reduction in cases [25].

There is a scarcity of studies investigating the impact of COVID-19 on suicidal behaviors that provide information on the characteristics of cases. As in our analysis, Yoshioka et al. observed a shift towards younger and female individuals [15]. However, to our knowledge, this is the first analysis of changes in other key features, such as social stressors (e.g., living alone, low socioeconomic status, among others) and underlying psychiatric diagnoses. Previous studies in our area found that mental health conditions such as schizophrenia, bipolar disorders, and personality as well as behavior disorders severely increased suicide risk [26]. Although complete suicide and suicidal behaviors should be taken as two separate entities [27,28], it is noteworthy that mental health comorbidities were less prevalent among individuals with suicidal behaviors after the onset of COVID-19. Taken together, our results suggest a shift towards lower risk profiles according to classical risk factors; lower prevalences of living alone and lacking social or family support are consistent with the growth in the youngest cases. Additionally, the increase in the rate of active suicidal ideation without an attempt was more pronounced than the increase in the rate of suicide attempts, likely reflecting the increase in suicidal behaviours in younger persons, who clinically express more ideation but fewer attempts. Whether the severity of the risk is actually lower cannot be confirmed due to a lack of data on suicide-related deaths. Finally, our analysis of individual characteristics before and after the onset of the pandemic showed that individuals in the low-socioeconomic-status group persistently account for the highest percentage of suicidal behaviors in our area. Although the data should be taken cautiously because, in contrast to age and gender, which remained constant throughout the analysis period, the population distribution across socioeconomic groups may have easily changed throughout the period due to the economic consequences of the pandemic, our findings suggest a shift towards less economically disadvantaged profiles. This trend is consistent with the incidence analysis. Although the incidence curve seems to increase more abruptly in the low-socioeconomic-status group, the relative increase was similar or even higher in the moderate-socioeconomic-status group (the low-socioeconomic-status group increased from a monthly incidence of 3.96 to 7.92 per 100,000 before the pandemic to 9.05 to 11.59 in the first semester of 2022; the corresponding increase for the moderate-socioeconomic-status group was from 1.35–3.36 to 4.09–5.43). 

Our study has several limitations that must be considered when interpreting its results. First, our findings should be constrained to the setting of suicidal behaviors, which may not necessarily reflect trend changes in suicide rate or changes in the characteristics of suicide completers during the pandemic. However, since previous suicidal behaviors are one of the strongest risk factors for death by suicide [26], our findings raise alarm. Second, some limitations associated with the study design should be considered, such as the uncertainty regarding the quality of reporting, typically seen in retrospective analyses. This is particularly relevant for the criteria of suicidal behavior and referral to a psychiatry ward following the activation of the Suicide Risk Code. Although the Suicide Risk Code program provides physicians with referral instructions [16], establishing suicidal behaviors may not be straightforward in some cases, and we cannot rule out heterogeneity between centers regarding these criteria. Third, owing to a change in the pharmaceutical co-payment criteria, we had to merge the low and very low socioeconomic statuses, thus losing sight of the differences between these two economically deprived profiles. Finally, under-reporting due to the overburdening of healthcare centers during the most severe waves of the outbreak, or the effects of perceived lower accessibility to healthcare services by the population, leading to not seeking medical help, should not be ruled out [29]. Despite these limitations, our analysis is strengthened by the population-based approach and the nationwide deployment of the Suicide Risk Code, which allowed for the collection of important information about episodes and individual states, including the suicidality module of the MINI scale.

## 5. Conclusions

Our population-based, specific assessment of suicidal behaviors in our area provides robust data on the alarming increase in suicidal behaviors within the years following the onset of the COVID-19 pandemic, which continues growing by the end of the investigated period. This increase is primarily explained by a higher incidence of suicidal behaviors among young women. Although we do not know whether these findings are also translated to suicide deaths in our area, the consistency of our results supports actions to be taken to revert this trend. It is worth highlighting that, owing to the younger profiles of the individuals committing suicide, the potential years of life lost due to suicide may overtake those due to COVID-19. The notable gender and socioeconomic effects, as well as the shift in the clinical profiles of individuals with suicidal behaviors, described in our study suggest that policies for preventing suicide in the pandemic and postpandemic contexts should focus on addressing these changes.

## Figures and Tables

**Figure 1 ijerph-19-14364-f001:**
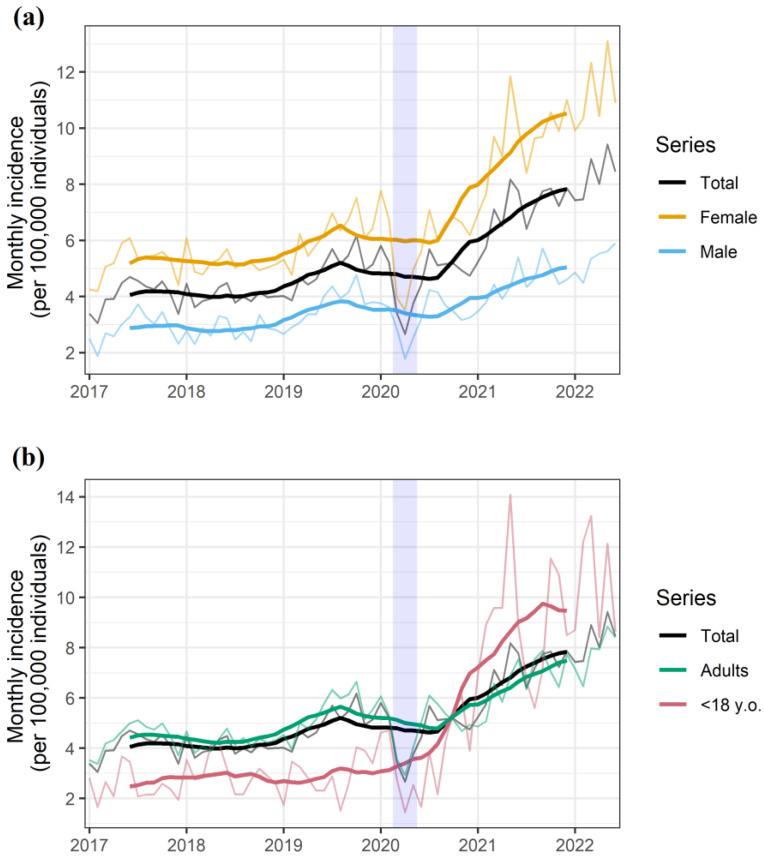
Monthly rate of suicidal behavior episodes for the overall population and stratified according to (**a**) gender as well as (**b**) age group. The shaded area represents the nationwide lockdown period, which was implemented on 13 March 2020, started a progressive easing on 10 May 2020, and was definitely lifted on 20 June 2020. The seasonal and random components of the observed rate are presented in Appendix A.

**Figure 2 ijerph-19-14364-f002:**
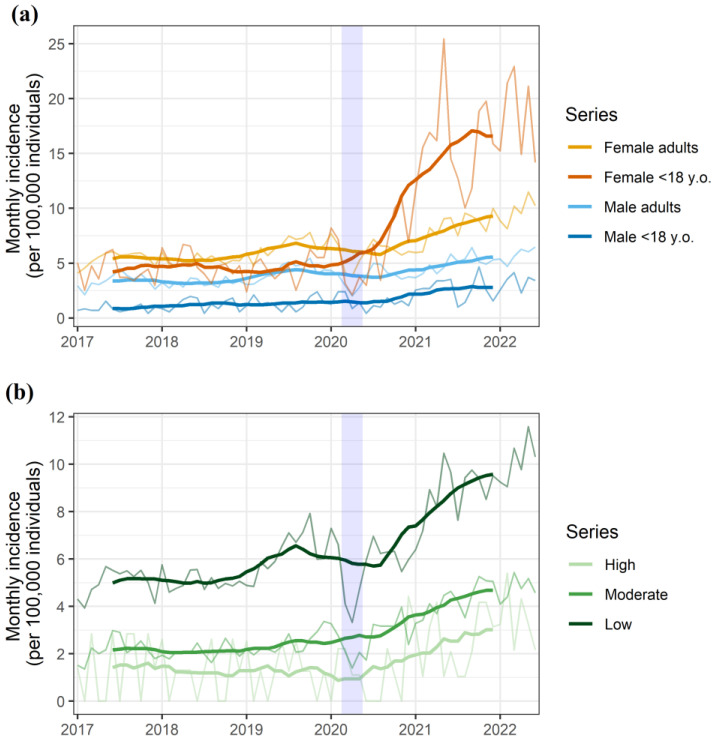
Monthly rate of suicidal behavior episodes for the overall population and stratified according to (**a**) gender and age group, as well as (**b**) socioeconomic status. The shaded area represents the nationwide lockdown period, which was implemented on 13 March 2020, started a progressive easing on 10 May 2020, and was definitely lifted on 20 June 2020.

**Figure 3 ijerph-19-14364-f003:**
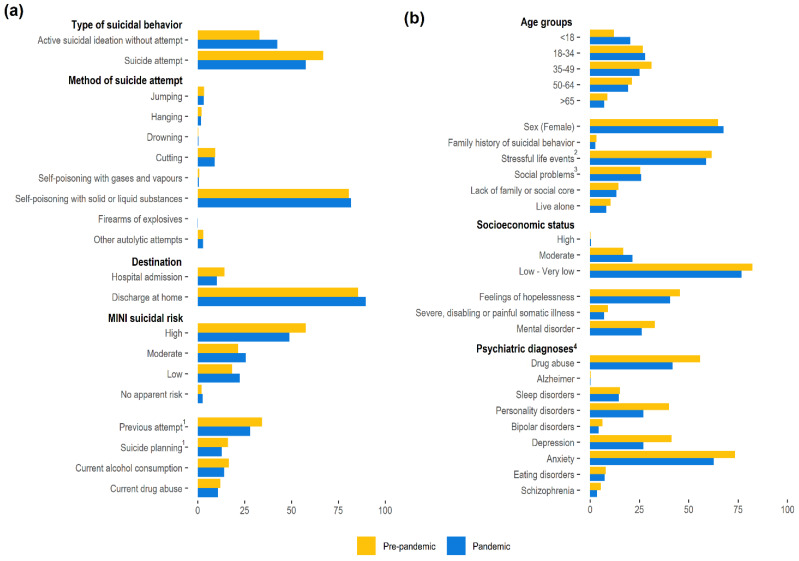
Characteristics of the episodes (**a**) and individuals (**b**) with suicidal behaviors. Values correspond to the average for the prepandemic period (i.e., 1 January 2017 to 29 February 2020) and pandemic period (i.e., 1 March 2020 to 30 June 2022). ^1^ As responded to the following items of the 6-item suicidality module of the Mini International Neuropsychiatric Interview (MINI): C4 (i.e., In the past months, did you have a suicide plan?) and C6 (i.e., In your lifetime, did you ever made a suicide attempt?). ^2^ Includes recent unemployment in addition to partner and family problems, among others. ^3^ Includes isolation, lack of support network, and socioeconomic difficulties. ^4^ Categories are not mutually exclusive. The category “drug abuse” includes any of the substances recorded (the percentages for each type of drug are listed in Appendix A). Depression includes recurrent or persistent mood (affective) disorders, and schizophrenia includes schizoaffective disorders.

**Table 1 ijerph-19-14364-t001:** Characteristics of the episode through the investigated period. Results are presented as no. and percentage of episodes.

	2017 (N = 3641)	2018 (N = 3638)	2019 (N = 4489)	2020 (N = 4291)	2021 (N = 6528)	2022 ^1^ (N = 3871)
**Type of Suicidal Behavior**						
Active suicidal ideation without attempt	1034 (28.4%)	1226 (33.7%)	1589 (35.4%)	1640 (38.22%)	2818 (43.17%)	1737 (44.87%)
Suicide attempt	2607 (71.6%)	2412 (66.3%)	2900 (64.6%)	2651 (61.78%)	3710 (56.83%)	2134 (55.13%)
**Suicide Attempt Method** ^2^						
Jumping	79 (3.03%)	88 (3.65%)	99 (3.41%)	111 (4.19%)	106 (2.86%)	54 (2.53%)
Hanging	47 (1.8%)	53 (2.2%)	70 (2.41%)	52 (1.96%)	63 (1.7%)	41 (1.92%)
Drowning	8 (0.31%)	8 (0.33%)	16 (0.55%)	13 (0.49%)	12 (0.32%)	15 (0.7%)
Cutting	232 (8.9%)	227 (9.41%)	284 (9.79%)	249 (9.39%)	352 (9.49%)	176 (8.25%)
Self-poisoning with gases and vapors	21 (0.81%)	15 (0.62%)	28 (0.97%)	25 (0.94%)	22 (0.59%)	7 (0.33%)
Self-poisoning with solid or liquid substances	2147 (82.36%)	1937 (80.31%)	2292 (79.03%)	2156 (81.33%)	3047 (82.13%)	1751 (82.05%)
Firearms or explosives	3 (0.12%)	4 (0.17%)	3 (0.1%)	0 (0%)	4 (0.11%)	2 (0.09%)
Other autolytic attempts	70 (2.69%)	80 (3.32%)	108 (3.72%)	45 (1.7%)	104 (2.8%)	88 (4.12%)
**Patient Destination**						
Hospital admission	2929 (80.44%)	3009 (82.71%)	4117 (91.71%)	3787 (88.25%)	5873 (89.97%)	3499 (90.39%)
Discharge at home	712 (19.56%)	629 (17.29%)	372 (8.29%)	504 (11.75%)	655 (10.03%)	372 (9.61%)
**MINI Suicidal Risk**						
High	2223 (61.05%)	2117 (58.19%)	2459 (54.78%)	2232 (52.02%)	3114 (47.7%)	1896 (48.98%)
Moderate	711 (19.53%)	843 (23.17%)	1006 (22.41%)	968 (22.56%)	1783 (27.31%)	988 (25.52%)
Low	623 (17.11%)	611 (16.79%)	919 (20.47%)	989 (23.05%)	1442 (22.09%)	872 (22.53%)
No apparent risk	84 (2.31%)	67 (1.84%)	105 (2.34%)	102 (2.38%)	189 (2.9%)	115 (2.97%)
**Previous attempt** ^3^	1298 (35.65%)	1327 (36.48%)	1431 (31.88%)	1332 (31.04%)	1806 (27.67%)	1013 (26.17%)
**Suicide planning** ^3^	686 (18.84%)	553 (15.2%)	674 (15.01%)	592 (13.8%)	810 (12.41%)	498 (12.86%)
**Current alcohol consumption**	640 (17.58%)	587 (16.14%)	756 (16.84%)	696 (16.22%)	879 (13.47%)	511 (13.2%)
**Current drug abuse**	442 (12.14%)	437 (12.01%)	550 (12.25%)	523 (12.19%)	698 (10.69%)	391 (10.1%)

^1^ The observation period for the year 2022 ends on 30 June 2022. ^2^ Percentages are over individuals with suicide attempt. ^3^ As responded to the following items of the 6-item suicidality module of the Mini International Neuropsychiatric Interview (MINI): C4 (i.e., In the past months, did you have a suicide plan?) and C6 (i.e., In your lifetime, did you ever made a suicide attempt?).

**Table 2 ijerph-19-14364-t002:** Sociodemographic and clinical characteristics of individuals with suicidal behaviors within the investigated period. Results are presented as no. and percentage of individuals.

	2017 (N = 3641)	2018 (N = 3638)	2019 (N = 4489)	2020 (N = 4291)	2021 (N = 6528)	2022 ^1^ (N = 3871)
**Sociodemographic Characteristics**						
Age Groups						
<18	412 (11.32%)	503 (13.83%)	473 (10.54%)	601 (14.01%)	1492 (22.86%)	866 (22.37%)
18–34	892 (24.5%)	928 (25.51%)	1294 (28.83%)	1227 (28.59%)	1768 (27.08%)	1131 (29.22%)
35–49	1194 (32.79%)	1118 (30.73%)	1387 (30.9%)	1215 (28.32%)	1608 (24.63%)	894 (23.09%)
40–64	807 (22.16%)	772 (21.22%)	934 (20.81%)	906 (21.11%)	1237 (18.95%)	695 (17.95%)
>65	336 (9.23%)	317 (8.71%)	401 (8.93%)	342 (7.97%)	423 (6.48%)	285 (7.36%)
Sex (female)	2371 (65.12%)	2394 (65.81%)	2851 (63.51%)	2801 (65.28%)	4483 (68.67%)	2655 (68.59%)
Family history of suicidal behavior	135 (3.71%)	119 (3.27%)	129 (2.87%)	124 (2.89%)	174 (2.67%)	92 (2.38%)
Stressful life events ^2^	2271 (62.37%)	2236 (61.46%)	2746 (61.17%)	2642 (61.57%)	3788 (58.03%)	2248 (58.07%)
Social problems ^3^	883 (24.25%)	848 (23.31%)	1239 (27.6%)	1144 (26.66%)	1691 (25.9%)	998 (25.78%)
Lack of family or social core	530 (14.56%)	530 (14.57%)	628 (13.99%)	597 (13.91%)	894 (13.69%)	489 (12.63%)
Living alone	408 (11.21%)	362 (9.95%)	459 (10.22%)	367 (8.55%)	542 (8.3%)	301 (7.78%)
Socioeconomic Level						
High	12 (0.33%)	11 (0.3%)	12 (0.27%)	13 (0.3%)	29 (0.44%)	19 (0.49%)
Moderate	634 (17.41%)	626 (17.21%)	711 (15.84%)	881 (20.53%)	1470 (22.52%)	790 (20.41%)
Low or very low	2972 (81.63%)	2982 (81.97%)	3741 (83.34%)	3353 (78.14%)	4970 (76.13%)	2983 (77.06%)
NA	23 (0.63%)	19 (0.52%)	25 (0.56%)	44 (1.03%)	59 (0.9%)	79 (2.04%)
**Clinical Characteristics**						
Feelings of hopelessness	1796 (49.33%)	1624 (44.64%)	1958 (43.62%)	1758 (40.97%)	2673 (40.95%)	1544 (39.89%)
Severe, disabling or painful somatic illness	365 (10.02%)	325 (8.93%)	405 (9.02%)	293 (6.83%)	456 (6.99%)	286 (7.39%)
Mental disorder	1239 (34.03%)	1165 (32.02%)	1462 (32.57%)	1256 (29.27%)	1721 (26.36%)	917 (23.69%)
Psychiatric history ^4^						
Drug abuse	2023 (55.56%)	2007 (55.17%)	2521 (56.16%)	2368 (55.19%)	2548 (39.03%)	1353 (34.95%)
Alzheimer	13 (0.36%)	11 (0.3%)	14 (0.31%)	12 (0.28%)	9 (0.14%)	9 (0.23%)
Sleep disorders	478 (13.13%)	500 (13.74%)	690 (15.37%)	662 (15.43%)	914 (14%)	522 (13.48%)
Personality disorders	1267 (34.8%)	1181 (32.46%)	1464 (32.61%)	1380 (32.16%)	1651 (25.29%)	794 (20.51%)
Bipolar disorders	221 (6.07%)	219 (6.02%)	245 (5.46%)	233 (5.43%)	258 (3.95%)	138 (3.56%)
Depression	1365 (37.49%)	1261 (34.66%)	1438 (32.03%)	1336 (31.13%)	1619 (24.8%)	819 (21.16%)
Anxiety	2394 (65.75%)	2421 (66.55%)	3168 (70.57%)	2982 (69.49%)	3932 (60.23%)	2126 (54.92%)
Eating disorders	272 (7.47%)	250 (6.87%)	338 (7.53%)	328 (7.64%)	490 (7.51%)	243 (6.28%)
Schizophrenia	209 (5.74%)	176 (4.84%)	215 (4.79%)	209 (4.87%)	219 (3.35%)	90 (2.32%)

^1^ The observation period for the year 2022 ends on 30 June 2022. ^2^ Includes recent unemployment in addition to partner and family problems, among others. ^3^ Includes isolation, lack of support network, and socioeconomic difficulties. ^4^ Categories are not mutually exclusive. The category “drugs” includes any of the substances recorded, including tobacco and alcohol; the percentages of each type of drug are presented in Appendix A. Depression includes recurrent or persistent mood (affective) disorders, and schizophrenia includes schizoaffective disorders.

## Data Availability

Data regarding suicidal behaviors are highly sensitive; therefore, raw data used in this analysis cannot be publicly available.

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
