# Peer review of "Impact of the COVID-19 Pandemic on the Incidence of Suicidal Behaviors: A Retrospective Analysis of Integrated Electronic Health Records in a Population of 7.5 Million"

_ijerph, 2022, doi:10.3390/ijerph192114364_

Round 1
Reviewer 1 Report
Brief Summary:
This retrospective, descriptive, analysis of electronic records, on a large subset of participants (N=21 920, out of 7.5 millions), aims to investigate the changes in suicidal behaviors and individual profile before and during the first two years of the COVID-19 pandemic. Considering the toll on mental health of the COVID-19 pandemic and the important worldwide disruption of lifestyles, this topic is very important and relevant. The current state of the COVID-19 pandemic is still uncertain as new waves may occur and therefore knowledge on its impact on mental health, especially suicidal behaviors, might bring new approaches to tackle this issue. Please see below my comments.
Introduction:
The first sentence of the introduction is unreferenced and might bring stigma to the Chinese community. Considering the vast amount of literature about COVID-19, it is suggested to open with a neutral, referenced, sentence as to ‘’what is SARS-CoV-2’’.
It is unclear what is meant by ‘’normal life’’ (lines 50-51) as this might differ vastly from one individual to another.
Considering the aim of this study, it is encouraged to add a paragraph on suicidal thoughts and behaviors. This topic itself is vast and important notions such as risk factors and protective factors of suicide should be included.
In the last paragraph, the aim of study is not clearly stated as well as the main hypothesis (Are lines 60-63 a hypothesis? If so, they could be included adjacent to a clearly formulated aim to account for clarity. Also, the current study focuses on the population of Catalonia, therefore, the studied shift in suicide should be about this population and not generalized.
In general, the introduction is clear and relevant to the topic.
Materials and methods:
Does the CHS contains the health information of all Catalonians, or does it include solely a subset (ex.: private versus public sector?). This should be clarified for the readership unfamiliar with Catalonia’s health policies.
Considering this study is on the population of Catalonia specifically, the second paragraph (except the last line) could be included and detailed in the introduction. This would account for readability of the section.
To better understand the Suicide Risk code dataset, it is suggested to include a table that defines the included variables. For instance, what is included in ‘’severe or painful systemic disease’’ or in ‘’lack of family or social network: is one friend a social network or are there specificities about this variable?’’.
Please include the validity and reliability of the MINI.
Line 132-133: it is unclear what is meant by ‘’moving average’’? Please define.
Line 133-134: what is meant by ‘’the seasonal component’’? Considering the readership is newly introduced to this term in your manuscript, please give a brief (one line) definition of what you mean by the analyzed components.
Results:
The results are concise and well presented.
Figure 1 is barely readable because of the number of graphs included. It is suggested to divide this Figure in two figures to account for readability.
Line 216: an ‘’SD’’ is missing before the 1.0.
In psychiatric diagnoses: schizophrenia is included. However, does it include other psychotic disorders than schizophrenia and schizoaffective disorder such as: schizophreniform disorder, delusion disorder, etc.? If so, this should be renamed.
Discussion:
The first sentence is misleading. The population studied is of 7.5 million, but the analysis is on 21 920 patients.
While the body of literature explored from lines 245 to 263 is extremely relevant to your discussion, ties with your own study are missing. How these findings compare to your results and why so? This should be further discussed.
Lines 269-271 in your study are very important. It would be interesting to discuss as to ‘’why’’ these mental health disorders are linked to an increase in suicidality during the pandemic and how does that link to your current study.
Limitations of your study are missing.
Conclusion:
Please outline ‘’why your study if over importance’’. It is believed that your manuscript is indeed relevant to the field. This should be better stated in the conclusion.
Overall appreciation of the manuscript:
This manuscript is very relevant to the field and highlights important profiles about suicidality that was derived from data pre and during the COVID-19 pandemic. The manuscript is well structured and clear. The method is aligned with the presented aim and the results and consistent with the methodology. The discussion should be further developed in relation to the findings of the study.
Minor comments:
Please be uniform as to how you reference articles. Sometimes (ex.: line 43, line 272) the reference appears in the middle of the sentence whereas sometimes it appears at the end.
Author Response
Brief Summary:
This retrospective, descriptive, analysis of electronic records, on a large subset of participants (N=21 920, out of 7.5 millions), aims to investigate the changes in suicidal behaviors and individual profile before and during the first two years of the COVID-19 pandemic. Considering the toll on mental health of the COVID-19 pandemic and the important worldwide disruption of lifestyles, this topic is very important and relevant. The current state of the COVID-19 pandemic is still uncertain as new waves may occur and therefore knowledge on its impact on mental health, especially suicidal behaviors, might bring new approaches to tackle this issue. Please see below my comments.
Response: We would like to thank the reviewer for highlighting the importance of the topic and providing insightful comments that have improved the accuracy of our report.
We have accepted most (if not all) of your suggestions and edited the manuscript accordingly. In the response, we indicate the lines corresponding to the tracked changed version. We hope the manuscript will be now suitable for publication in the IJEPH.
Introduction:
The first sentence of the introduction is unreferenced and might bring stigma to the Chinese community. Considering the vast amount of literature about COVID-19, it is suggested to open with a neutral, referenced, sentence as to ‘’what is SARS-CoV-2’’.
Response: We agree that the previous phrasing may lead to stigma to Chinese community and that this information is irrelevant for the purpose of the sentence. In the revised version of the manuscript, we have reworded it (lines 38-41)
It is unclear what is meant by ‘’normal life’’ (lines 50-51) as this might differ vastly from one individual to another.
Response: We have rephrased the sentence to avoid the term “normal life”, which has been used extensively in the press setting but may not be standardized in the scientific environment (lines 51-53)
Considering the aim of this study, it is encouraged to add a paragraph on suicidal thoughts and behaviors. This topic itself is vast and important notions such as risk factors and protective factors of suicide should be included.
Response: We agree that information in this regard will be helpful for the readers. We have included this in the introduction section (Lines 67-69)
In the last paragraph, the aim of study is not clearly stated as well as the main hypothesis (Are lines 60-63 a hypothesis? If so, they could be included adjacent to a clearly formulated aim to account for clarity. Also, the current study focuses on the population of Catalonia, therefore, the studied shift in suicide should be about this population and not generalized.
Response: We thank the reviewer for highlighting that the study hypothesis and objective is not clear enough. In the revised manuscript, we have changed the last paragraph of the introduction section to make the hypothesis and study objectives more clear (lines 73-75)
In general, the introduction is clear and relevant to the topic.
Materials and methods:
Does the CHS contains the health information of all Catalonians, or does it include solely a subset (ex.: private versus public sector?). This should be clarified for the readership unfamiliar with Catalonia’s health policies.
Response: We agree that the healthcare system and structure is heterogeneous across countries, and that the public/private coverage of healthcare services is mainstay for understanding the generalizability of the results. Although some private centres exist in Catalonia, these are rare, particularly in the emergency setting, where episodes of suicidal behaviours are primarily reported. In the revised version of the manuscript, we have re-phrased lines 80-83 of the Material and Methods section to provide this information.
Considering this study is on the population of Catalonia specifically, the second paragraph (except the last line) could be included and detailed in the introduction. This would account for readability of the section.
Respnse: We agree that providing the geographical setting in the introduction section will provide the reader with an idea of the external validity of our analysis. However, we think that providing all details of this paragraph in the introduction may make the reader lose sight on the study rationale. Therefore, in the revised manuscript, we have kept the second paragraph of the the “Study design and setting” (as recommended by the RECORD guidelines for retrospective analyses of administrative datasets) and included a succinct summary of the Suicide Risk Code program and the geographical setting in the last paragraph of the introduction section (lines 78-80)
To better understand the Suicide Risk code dataset, it is suggested to include a table that defines the included variables. For instance, what is included in ‘’severe or painful systemic disease’’ or in ‘’lack of family or social network: is one friend a social network or are there specificities about this variable?’’.
Response: We agree with the reviewer’s opinion that a table with detailed definitions of the parameters would be of utmost utility. As suggested, we have included a supplementary table with a brief description of all variables collected.
Please include the validity and reliability of the MINI.
Response: in the previous version of the manuscript, we cited the work by Sheehan et al. In the revised version, we have included details regarding the validity of the MINI scale (lines 132-134)
Line 132-133: it is unclear what is meant by ‘’moving average’’? Please define.
Response: as suggested, we have included a brief definition of the moving average (lines 156-159)
Line 133-134: what is meant by ‘’the seasonal component’’? Considering the readership is newly introduced to this term in your manuscript, please give a brief (one line) definition of what you mean by the analyzed components.
Response: we agree that more statistical details should be provided for the readers to better understand the methodological approach for the trend. In the revised version of the manuscript, we have expanded the description of this approach (lines 160-162).
Results:
The results are concise and well presented.
Figure 1 is barely readable because of the number of graphs included. It is suggested to divide this Figure in two figures to account for readability.
Response: As suggested, we have split Figure 1 into two figures so that all components are bigger and more clear. We believe the new distribution of panels is now more readable.
Line 216: an ‘’SD’’ is missing before the 1.0.
Response: added
In psychiatric diagnoses: schizophrenia is included. However, does it include other psychotic disorders than schizophrenia and schizoaffective disorder such as: schizophreniform disorder, delusion disorder, etc.? If so, this should be renamed.
Response: In psychiatric diagnosis, we have included as “schizophrenia” only the codes in F20* (schizophrenia) and F25*(schizoaffective disorders), as per the International classification of diseases 10-CM. This has been quoted in table S1 in supplementary materials.
Discussion:
The first sentence is misleading. The population studied is of 7.5 million, but the analysis is on 21 920 patients.
Response: In this sentence, we wanted to place the study in context by mentioning the source population (as defined in the RECORD statement). However, we agree that, without the study population, the sentence was misleading. In the revised version, we have added that 21,920 cases were identified in the source population of 7.5-milion people (lines 281-283)
While the body of literature explored from lines 245 to 263 is extremely relevant to your discussion, ties with your own study are missing. How these findings compare to your results and why so? This should be further discussed.
Response: We have re-phrased the second paragraph of the discussion section to better nail the literature with our findings (lines 290-310)
Lines 269-271 in your study are very important. It would be interesting to discuss as to ‘’why’’ these mental health disorders are linked to an increase in suicidality during the pandemic and how does that link to your current study.
Response: The link between some mental health disorders may slightly differ between countries and cultures, but is overall accepted in the field. In the revised manuscript, we have included two sentences in the introduction section in this regard (lines 67-70). Also, as the reviewer suggests, we have expanded and reworded the interpretation of our results in this regard in the discussion section (lines 324-329). In this interpretation, we have been careful to avoiding too speculative interpretations regarding the reason for these risk factors to lose weight due to the pandemic. However, we believe that in the current version, information and discussion in this regard is more complete.
Limitations of your study are missing.
Response: the paragraph before the conclusions section was intended to highlight the limitations of our analysis. In the revised version of the manuscript, we have identified them as limitations explicitly. We have also added the limitation regarding the need for merging the two lowest socioeconomic levels, which precluded from investigating differences between the low and very low socioeconomic profiles (lines 345-358)
Conclusion:
Please outline ‘’why your study if over importance’’. It is believed that your manuscript is indeed relevant to the field. This should be better stated in the conclusion.
Response: we thank the reviewer for encouraging us to highlight the relevance of our analysis. In the revised version, we have expanded the conclusions section by identifying some of the strengths of our study (lines 367-378)
Overall appreciation of the manuscript:
This manuscript is very relevant to the field and highlights important profiles about suicidality that was derived from data pre and during the COVID-19 pandemic. The manuscript is well structured and clear. The method is aligned with the presented aim and the results and consistent with the methodology. The discussion should be further developed in relation to the findings of the study.
Minor comments:
Please be uniform as to how you reference articles. Sometimes (ex.: line 43, line 272) the reference appears in the middle of the sentence whereas sometimes it appears at the end.
Response: we have revised the manuscript in this regard and tried to be as consistent as possible. Please, notice that the criteria was to place the reference as close as possible to the specific claim or information piece. Therefore, in sentences that provide more than one piece of information, reference may appear separated.

Reviewer 2 Report
This manuscript highlights the impact of COVID-19 pandemic on the suicidal behaviors. Commendations to the authors for taking on this serious issue during the pandemic. The manuscript is very well-written and adheres to the journal’s guidelines. Nonetheless, it can be improved using the below notes.
1. Line 95-100: It would be better to have a separate section for details regarding ethical approval.
2. Kindly provide the reference no. of the ethical approval.
3. It would be good to add some inferential statistics [e.g. comparing socio-demographic and clinical characteristics before and during COVID-19 pandemic, and among suicidal behavior groups (active suicidal ideation without attempt vs suicide attempts), comparing the type of suicide behaviors before and during the pandemic etc.]
4. Remove reference “[11]” from the conclusion.
Author Response
Comments and Suggestions for Authors
This manuscript highlights the impact of COVID-19 pandemic on the suicidal behaviors. Commendations to the authors for taking on this serious issue during the pandemic. The manuscript is very well-written and adheres to the journal’s guidelines. Nonetheless, it can be improved using the below notes.
Response: We thank the reviewer for his/her interest in our work and the insightful comments. We have accepted most of your suggestions and edited the manuscript accordingly. We hope the manuscript will be now suitable for publication in the IJEPH.
- Line 95-100: It would be better to have a separate section for details regarding ethical approval.
Response: A new sub-section has been created, as suggested.
- Kindly provide the reference no. of the ethical approval.
Response: the reference approval and date have been provided (line 117)
- It would be good to add some inferential statistics [e.g. comparing socio-demographic and clinical characteristics before and during COVID-19 pandemic, and among suicidal behavior groups (active suicidal ideation without attempt vs suicide attempts), comparing the type of suicide behaviors before and during the pandemic etc.]
Response: While inferential statistics are common in comparisons between groups, two factors make them controversial (or even inappropriate) in our analysis. First, it was a population-based approach. Hypothesis tests are based on the principle of sampling, which was not done in our analysis owing to the population-based approach. Second, the primary endpoint (and all analyses except panel b in figure 3) referred to episodes rather than individuals, thus increasing the risk of biases and misinterpretations of these analyses. Since the question raised by the reviewers may also be a concern of a reader, we have added a sentence in the statistical analysis section to justify the lack of inference analyses.
- Remove reference “[11]” from the conclusion.
Response: the reference has been removed, as suggested.
Reviewer 3 Report
Please see the attached.

Author Response
This study presents important findings and valuable contributions to understanding the impact of COVID-19 on the incidence of suicidal behaviors. However, the following issues require attention:
Response: We thank the reviewer for appraising the contribution of our results to the field and for his/her insightful comments on our manuscript.
We have accepted most (if not all) of your suggestions and edited the manuscript accordingly. In the response, we indicate the lines corresponding to the tracked changed version. We hope the manuscript will be now suitable for publication in the IJEPH.
- Since this manuscript examine the impact of COVID-19 pandemic on the incidence of suicidal behaviors in Catalonia (Spain) population, it is important to provide background information on the topic including Catalonia’s lockdown period, mental health care utilization rates, and the effect of COVID-19.
Response: We agree that the background situation of Catalonia may help the reader in interpretation and generalizability of the results. In the revised manuscript, we have added information regarding the course of the pandemic in Catalonia (lines 91-99) and provided supplementary data on healthcare resource utilization.
- Please mention the use of secondary data in the abstract.
Response: we have mentioned it in the abstract, as suggested (line 23)
- Did the authors separately collect data on suicide risk levels? Also, please provide a rationale for the cutoff score for the suicidality score.
Response: The suicide risk level was used as established by the suicidality module of the MINI scale, using the cut-off levels defined in the validation studies of this scale. For more clarity, we have added the reference at the end of the sentence (line 136).
- Some of the discussion is the repetition of the results with little discussion, please expand the discussion. Also, study results show that the rate of active suicidal ideation without attempt was increased since 2017, while the suicide attempt rate was decreased. The explanation in this regard will also be of interest to the readers.
Response: As suggested, we have rephrased some parts of the discussion section to better place our results in the context of the existing literature. Also, we have tried to clarify the interpretation regarding the trends of suicidal ideation without attempts and those with attempts.
Round 2
Reviewer 1 Report
The authors have addressed all of my comments.
Reviewer 3 Report
The revised paper is much improved.